# Comparison of Personal or Indoor PM_2.5_ Exposure Level to That of Outdoor: Over Four Seasons in Selected Urban, Industrial, and Rural Areas of South Korea: (K-IOP Study)

**DOI:** 10.3390/ijerph20176684

**Published:** 2023-08-30

**Authors:** Chiyou Song, Chris Chaeha Lim, Birhan Legese Gurmu, Mingi Kim, Sangoon Lee, Jinsoo Park, Sungroul Kim

**Affiliations:** 1Department of ICT Environmental Health System, Graduate School, Soonchunhyang University, Asan 31538, Republic of Korea; chiyou1997@gmail.com (C.S.);; 2Department of Community, Environment and Policy, Zuckerman College of Public Health, University of Arizona, Tucson, AZ 85724, USA; chrislim@arizona.edu; 3Department of Environmental Health Sciences, Graduate School, Soonchunhyang University, Asan 31538, Republic of Korea

**Keywords:** personal exposure, PM_2.5_, indoor, outdoor

## Abstract

This study aimed to compare the distribution of indoor, outdoor, and personal PM_2.5_ (particulate matter ≤ 2.5 μm) hourly concentrations measured simultaneously among 81 nonsmoking elderly participants (65 years or older) living in urban, industrial, or rural areas over 4 seasons (2 weeks per season) from November 2021 to July 2022). PM_2.5_ measurements were conducted using low-cost sensors with quality control and quality assurance tests. Seasonal outdoor PM_2.5_ levels were 16.4 (9.1–29.6) μg/m^3^, 20.5 (13.0–38.0) μg/m^3^, 18.2 (10.2–31.8) μg/m^3^, and 9.5 (3.8–18.7) μg/m^3^ for fall, winter, spring, and summer, respectively. For indoor PM_2.5_, the median seasonal range was 5.9–7.5 μg/m^3^, and the median personal PM_2.5_ exposure concentration was 8.0–9.4 μg/m^3^. This study provided seasonal distributions of IO (ratio of indoor to outdoor PM_2.5_ concentration) and PO (ratio of personal to outdoor PM_2.5_ concentration) using a total of 94,676 paired data points. The median seasonal IO ranged from 0.30 to 0.51 in fall, winter, and spring; its value of summer was 0.70. The median PO by season and study area were close to 1.0 in summer while it ranged 0.5 to 0.7 in other seasons, statistically significantly lower (*p* < 0.05) than that in summer. Our study has revealed that the real-world exposure level to PM_2.5_ among our elderly study participants might be lower than what was initially expected based on the outdoor data for most of the time. Further investigation may need to identify the reasons for the discrepancy, personal behavior patterns, and the effectiveness of any indoor air quality control system.

## 1. Introduction

National monitoring sites use stationary monitors to measure levels of particulate matter less than 2.5 μm (PM_2.5_) in ambient air. These monitors are strategically placed and provide representative measurements for a particular area [1,2]. However, they may not capture the microscale variations in the PM_2.5_ concentrations that individuals are exposed to throughout the day [3,4,5]. Personal exposure to PM_2.5_ can be influenced by factors such as proximity to major roads, industrial emissions, indoor air quality, ratio of time spent outdoors, and weather conditions [4]. For example, higher levels of PM_2.5_ could be measured from individuals living near busy highways or industrial areas compared with measurements taken at a national monitoring site located further away.

Moreover, indoor environments can have different PM_2.5_ levels compared with outdoor environments. Indoor sources of PM_2.5_ such as cooking fumes, tobacco smoke, and certain building materials can contribute to elevated PM_2.5_. Therefore, personal exposure levels can vary significantly depending on the time spent indoors versus outdoors.

Thus, an increasing number of studies focus on indoor air pollution levels [6]. In 2021, with consideration of exposure pathways, the WHO risk assessment guidebook even emphasized the use of both indoor and outdoor measurement data and time–activity patterns for improved risk assessment of air pollutants [6].

Owing to technological development, researchers often use portable personal air quality monitors or sensors to obtain accurate assessments of personal exposure to PM_2.5_. These devices are worn by individuals and provide real-time measurements of PM_2.5_ in the immediate vicinity [7,8,9]. This approach allows for a more precise understanding of the actual exposure that individuals experience during their daily activities [5]. Since the late 2010s, personal or portable particulate matter measurement sensors have been introduced, and these devices demonstrated reliable accuracy and precision levels, compared with outcomes from national monitoring devices [10]. This allowed for improvements in the estimation of personal exposure.

However, research is lacking on personal PM_2.5_ exposure levels relative to outdoor and indoor concentrations. In particular, data on the elderly, which are considered one of the most vulnerable groups to PM_2.5_, are extremely scarce both domestically and internationally.

A previous study conducted with the Korean elderly concluded that exposure to airborne PM_2.5_ was associated with cognitive decline [11], cataract development in old age [12], and myocardial infarction [13]. However, few studies have been conducted on personal exposure and health effects in the elderly.

The purpose of this study was to measure real-time, sensor-based personal PM_2.5_ concentrations as well as the indoor and outdoor PM_2.5_ concentrations for elderly participants and to evaluate the associations between personal and indoor PM_2.5_ or between personal and outdoor PM_2.5_. Ultimately, we aimed to understand the margin of discrepancy of PM_2.5_ exposure levels among indoor, outdoor, and personal monitoring with the ratio of personal to outdoor (PO) or indoor to outdoor (IO) PM_2.5_.

## 2. Materials and Methods

### 2.1. Study Participants

This study included 81 nonsmoking elderly people aged 65 years or older living in urban, industrial, or rural areas and was conducted during fall, winter, spring, and summer from November 2021 to July 2022. PM_2.5_ concentration was measured by installing indoor, outdoor, and personal particulate matter sensors on the subjects for two weeks per season.

Participants were recruited mainly from those enrolled in the Korean Genome and Epidemiology Study (KoGES) at Korea University Ansan Hospital and in KFACS at Kyung Hee University Seoul Hospital and Korea University Guro Hospital. Some participants in KFACS and those in rural areas were recruited among elders joining the epidemiology study in the Chungnam Thermal Power Plant district (Figure 1).

The study participants were older individuals who were able to install indoor and outdoor PM_2.5_ sensors in their homes. We selected volunteers who could install a multi-tap sensor outside their houses or on verandas or terraces that allow adequate ambient air circulation. Simultaneously, the indoor sensors were installed in living rooms at a height of approximately 1.5 m from the floor. Field managers visited participants’ houses at least twice per season (at the start and end of seasonal sampling or when requested). We paid compensation fees each season at the end of seasonal sampling. The fees were paid four times in total. Among those who agreed, we used a smartphone to measure personal PM_2.5_ and operate sensors as well as smart bands to measure their heartbeat.

The consent form and study plans were reviewed and approved by the Institutional Review Board of Soonchunhyang University (202105-SB-054-2). KFACS added a subcohort of PM_2.5_ and received research approval from the Research Ethics Committee of Kyung Hee University Hospital (KHUH 2021-05-081-009). KoGES received research approval from the Research Ethics Committee of Ansan Hospital of Korea University (2006AS0045).

### 2.2. Outdoor, Indoor, and Personal PM_2.5_ Measurements

#### PM_2.5_ Measurement Methods

Indoor, outdoor, and personal PM_2.5_ exposure levels were measured simultaneously and in conjunction with the outdoor temperature, humidity, pressure (outdoor sensor: PurpleAir, Draper, UT, USA), indoor temperature, humidity, CO_2_ (indoor sensor: Awair, San Francisco, CA, USA), and latitude and longitude (personal sensor: PICO, Seoul, Republic of Korea).

Measurements were taken for two weeks per season: fall (24 November 2021 to 7 December 2021), winter (11 February 2022 to 24 February 2022), spring (25 March 2022 to 7 April 2022), and summer (21 June 2022 to 4 July 2022). The indoor, outdoor, and personal PM_2.5_ sensor devices used in this study were certified as class 1 with an error rate of less than 20% in comparison with the national reference methods system [14] and the guidelines of the US Environmental Protection Agency [15].

Each indoor sensor was installed in the living room at a height of approximately 1.5 m from the floor. It was connected to Wi-Fi and conducted measurements every 5 min. Each outdoor sensor was installed on an outside safety bar on the veranda of the subjects’ homes, and it was securely fixed with cable ties and conducted measurements every 2 min. Participants who agreed to personal sampling carried the personal sensor in a carrying pouch attached to the participant’s waist belt or bag, and measurements were obtained at one-minute intervals over two weeks per season or the longest period spanning at least five consecutive days.

A smart band was provided to the volunteers along with instruction guidelines for charging. All data were sent to the ‘ESCORTandCARE’ data servers via a smartphone application or API.

For the internal quality control and quality assurance of PM_2.5_ sensors, the instrumental error rate was evaluated by referring to our previous study [16] and the Quality Assurance Handbook for Air Pollution Measurement Systems by the US Environmental Protection Agency in 2017 [15].

For this evaluation, 15 sensors out of each of the 81 outdoor, indoor, and personal sensors were randomly selected and compared with a reference device (GRIMM, Hamburg, Germany) twice: before the start of this study (October 2021) and end of this study (August 2022) (Appendix A, Appendix A). A sensor was considered suitable for use when it demonstrated an accuracy of more than 80% compared with the reference [16].

### 2.3. Data Preparation

To compare the real-time PM_2.5_ concentrations (outdoor: every two minutes; indoor: every five minutes; and personal: every minute) with those at the national monitoring site, we first compiled a database of the hourly mean outdoor, indoor, and personal sensor measurements according to time and place. Outdoor, indoor, and personal PM_2.5_ data were then merged with the national PM_2.5_ measurement data on the basis of the participant’s residential address. National data were downloaded from the official data site of AIR KOREA (http://m.airkorea.or.kr/main, accessed on 20 November 2022).

Based on the residential addresses of the 81 elderly participants, 19 national measurement stations located in Seoul, Incheon metropolitan city and the Gyeonggi province were used for subjects living in cities; a total of 7 national measurement networks located in Ansan City were used for subjects living in industrial complexes; and 13 national measurement networks located in Dangjin City, Boryeong City, Taean County, and Buyeo County were used for subjects living in rural areas. To match with high-resolution outdoor PM_2.5_ data, we also used temperature, relative humidity, and air pressure obtained from outdoor sensors.

To match the real-time sensor data and national measurement data, personal, outdoor, and indoor sensor data were matched by time, regional location, and participant’s id. The diary information on temporal activity patterns (1. sleeping at home; 2. resting at home; 3. socializing outside the community center; 4. socializing outside shops/restaurants; 5. exercise; 6. cleaning indoors; 7. farming outdoors; 8. cooking indoors; 9. working outdoors as a part-time job) collected during the measurement period was also linked to the hourly data. Hourly PM_2.5_ data from the national site were merged into the database of indoor, outdoor, and personal PM_2.5_ concentrations, and the ratios of indoor, outdoor, and personal levels to national levels were calculated (Figure 2). Finally, the ratio of personal to outdoor PM_2.5_ levels for each period was used to derive unknown personal exposure estimates.

The hourly national, indoor, outdoor, and personal PM_2.5_ data were screened by Tukey’s outlier test to remove outliers (Figure 2).

Outdoor PM_2.5_ values were then excluded from the analysis when outdoor barometric pressure was 950 hPa or outdoor temperature was above 50 °C. If the value of outdoor relative humidity (%) was 101 or larger, we considered the data to be unreliable and excluded them from analysis. The sensors installed outside the participants’ homes in each study area were then paired with data from the national measurement data by using the shortest distance based on latitude and longitude.

Finally, 94,676 data pairs were collected at 1 h intervals daily for 2 weeks in each of the 4 seasons. These data pairs were derived from 81 participants who had outdoor measurements in winter, spring, and summer in this study.

## 3. Results

### 3.1. Distribution of PM_2.5_ Concentrations and Meteorological Conditions

For outdoor PM_2.5_ measured at national monitoring sites, the median (interquartile range (IQR) was 16.0 (9.0–27.0) μg/m^3^ when using all data without seasonal separation and 17.0 (10.0–28.0) μg/m^3^, 20.0 (13.0–28.0) μg/m^3^, and 16.0 (10.0–27.0) μg/m^3^ for fall, winter, and spring, respectively (Table 1). In summer, the median was 8.0 (4.0–17.0) μg/m^3^ and was statistically significantly different from those of other seasons (*p* < 0.001). The overall outdoor PM_2.5_ level by real-time sensors was 16.3 (9.1–29.4) μg/m^3^. Seasonal outdoor PM_2.5_ levels were 16.4 (9.1–29.6) μg/m^3^, 20.5 (13.0–38.0) μg/m^3^, 18.2 (10.2–31.8) μg/m^3^, and 9.5 (3.8–18.7) μg/m^3^ for fall, winter, spring, and summer. Distribution of outdoor sensor-based PM_2.5_ data was similar to that of the national air quality data.

For indoor PM_2.5_, the median seasonal range was 5.9–7.5 μg/m^3^, and the median personal PM_2.5_ exposure concentration was 8.0–9.4 μg/m^3^. The seasonal distribution of indoor or personal PM_2.5_ concentrations was also lower in summer than in the other seasons, and there was a statistically significant difference between the seasons (*p* < 0.001).

The medians (IQR) of the outdoor temperature and humidity were the lowest in winter at 5.0 °C (0.7–10.8 °C) and 32.4% (25.0–40.4%), respectively, and the highest in summer at 30.5 °C (28.6–32.6 °C) and 57.2% (50.4–63.8%), respectively; these values reflected the characteristics of the seasons. The indoor temperatures in fall, winter, and spring were 20.3 °C (17.9 –22.4 °C), 21.2 °C (18.0–23.7 °C), and 20.9 °C (18.6–23.3 °C), respectively, and the indoor humidity was 43.1% (35.4–51.0%), 31.8% (24.2–43.4%), and 46.9% (38.6–56.2%). In summer, the indoor temperature and relative humidity levels were 27.6 °C (26.4–28.6 °C) and 73.8% (67.7–78.8%), respectively, which were significantly different from those recorded outdoors [temperature: 30.5 °C (28.6–32.6 °C) versus relative humidity: 57.2% (50.4–63.8%)]. Additionally, indoor temperature and humidity levels in summer were significantly different from those in other seasons. The outdoor barometric pressure showed statistically significant differences between seasons (*p* < 0.001). In addition, the distributions of most indoor, and personal PM_2.5_ concentrations investigated in this study in each season were different, with the outdoor PM_2.5_ concentrations being higher than the indoor and personal PM_2.5_ concentrations.

### 3.2. Indoor Outdoor Personal PM_2.5_ Concentration Distributions by Season and Region

Figure 3 shows the distributions of outdoor, indoor, and personal PM_2.5_ concentrations and meteorological condition by season and region (urban, industrial, and rural area).

The median (IQR) outdoor PM_2.5_ concentration in fall was 21.4 (12.3–38.5) μg/m^3^ in industrial areas, 16.3 (8.7–31.3) μg/m^3^ in rural areas, and 13.5 (7.8–23.9) μg/m^3^ in urban areas, whereas the median (IQR) national PM_2.5_ concentration was 18.0 (11.0–33.0) μg/m^3^ in industrial complexes, 17.0 (11.0–28.0) μg/m^3^ in cities, and 15.0 (9.0–25.0) μg/m^3^ in rural areas. Median indoor and personal PM_2.5_ concentrations ranged from 5.9 to 6.5 μg/m^3^ and 8.4 to 9.2 μg/m^3^, respectively, and there were regional differences (*p* < 0.001) in national network, indoor, outdoor, and personal PM_2.5_ concentrations in fall.

In winter, the median (IQR) outdoor PM_2.5_ concentration was the highest in industrialized areas at 23.1 (14.6–41.4) μg/m^3^; the median in rural and urban areas was 19.7 (12.1–44.5) μg/m^3^ and 19.6 (12.6–32.6) μg/m^3^, respectively. The median (IQR) outdoor PM_2.5_ in the national measurement network was the highest in industrial areas at 20.0 (13.0–38.0) μg/m^3^, followed by cities at 19.0 (13.0–32.0) μg/m^3^, and industrial areas at 19.0 (13.0–40.0) μg/m^3^. The median (quartile) indoor PM_2.5_ concentration was 7.5 (4.4–13.5) μg/m^3^ in urban areas, 7.3 (4.8–12.6) μg/m^3^ in industrial areas, and 7.8 (4.5–15.7) μg/m^3^ in rural areas. The median (IQR) personal PM_2.5_ concentration was 10.1 (7.3–18.3) μg/m^3^ in rural areas, 9.4 (6.7–16.4) μg/m^3^ in industrial zones, and 8.8 (6.1–14.5) μg/m^3^ in urban areas. There were statistical regional differences between the national measurement network and outdoor personal PM_2.5_ concentration (*p* < 0.001).

In spring, the median (quartile) outdoor PM_2.5_ concentration was 20.9 (11.8–36.8) μg/m^3^ in industrial zones, 18.8 (11.3–30.2) μg/m^3^ in rural areas, and 15.4 (8.7–29.5) μg/m^3^ in urban areas. The median (quartile) national PM_2. 5_ concentration was the highest in industrialized areas at 18.0 (12.0–30.0) μg/m^3^, and those in urban and rural areas were 16.0 (10.0–28.0) μg/m^3^ and 16.0 (10.0–24.0) μg/m^3^, respectively. The median indoor personal PM_2.5_ concentration was between 6.6 and 7.4 μg/m^3^, with statistically significant regional differences between the national network, outdoor, indoor, and personal PM_2.5_ concentrations (*p* < 0.001).

In summer, the median (IQR) outdoor PM_2.5_ concentration was the lowest in rural areas at 8.1 (2.9–16.3) μg/m^3^, and those in urban and industrial areas were 10.1 (4.2–19.2) μg/m^3^ and 10.5 (4.2–20.5) μg/m^3^, respectively. The median (quartile) national PM_2.5_ concentrations was 10.0 (5.0–20.0) μg/m^3^ in urban areas, 8.0 (4.0–18.0) μg/m^3^ in industrial areas, and 6.0 (3.0–12.0) μg/m^3^ in rural areas. The median (IQR) indoor PM_2.5_ concentration was 6.5 (3.8–11.1) μg/m^3^ in industrial zones, 6.0 (3.2–10.8) μg/m^3^ in urban areas, and 5.4 (3.3–9.0) μg/m^3^ in rural areas. The median personal PM_2.5_ concentration was 8.8 (5.9–16.1) μg/m^3^ in urban areas, 7.9 (5.8–15.1) μg/m^3^ in industrial complexes, and 7.1 (5.6–12.3) μg/m^3^ in rural areas. There were statistically significant differences between the national measurement network, indoor, outdoor, and personal PM_2.5_ concentrations by region (*p* < 0.001).

As we mentioned earlier, there was a significant difference in meteorological conditions among seasons. Furthermore, we found a significant difference among study areas with each season. In winter, the median (IQR) outdoor and indoor temperatures were higher in urban areas at 8.5 °C (2.3–14.4 °C) and 22.9 °C (20.7–24.6 °C), respectively, and lower in industrial areas at 3.0 °C (0.6–7.0 °C) and 20.1. °C (18.6–22.8 °C), respectively. In summer, the median (IQR) outdoor and indoor temperatures were higher in urban areas at 31.6 °C (29.9 °C–33.3 °C) and 27.8 °C (26.9 °C–29.0 °C), respectively, and lower in rural areas at 29.3 °C (27.6 °C–31.6 °C) and 27.0 °C (25.8 °C–28.3 °C), respectively. In summer, outdoor humidity was highest in rural areas at 62.9% (54.6–69.0%) and lowest in urban areas at 53.8% (47.9–58.7%). Meteorological conditions as well as outdoor, indoor, and personal PM_2.5_ concentrations all differed between regions (*p* < 0.001).

### 3.3. Distribution of the Ratios of IO and PO by Season and Area

The data obtained by season were further separated to compare the IO (ratio of indoor to outdoor) and PO (ratio of personal to outdoor PM_2.5_) ratios in the three urban–industrial–rural regions during the same season (Table 2).

The median (IQR) of IO in winter was 0.42 (0.24–0.63) in urban areas; 0.34 (0.24–0.43) in industrialized areas; 0.36 (0.26–0.53) in rural areas. That of PO was 0.50 (0.33–0.68) and 0.46 (0.33–0.62) in urban and industrialized areas, respectively; and 0.54 (0.39–0.73) in rural areas. In summer, the medians of IO ranged 0.68 to 0.72 in urban, industry. and rural area. The median (IQR) for PO was 1.11 (0.84–1.67) for urban areas, 1.07 (0.784–1.57) for industrial areas and 1.12 (0.75–2.06) for rural areas. Unlike in the other seasons, median POs were above 1.00 in all regions, and IO and PO in summer were statistically different among regions (*p* < 0.001).

## 4. Discussion

Our study compared outdoor, indoor, and personal hourly PM_2.5_ concentrations measured by light scattering sensors over four seasons and three typical study areas (urban industrial and rural area) together with the national ambient PM_2.5_ measurement data.

In this study, the outdoor real-time PM_2.5_ concentration 16.3 (9.1–29.4) µg/m^3^ was similar to that of the national monitoring site results at 16.0 (9.0–27.0) µg/m^3^. The median value of real-time outdoor PM_2.5_ levels in summer was 9.5 µg/m^3^, which was 2 or 2.5 times lower than the one in other seasons. Outdoor PM_2.5_ was highest in winter indicating that there was potential seasonal difference of ambient PM_2.5_ levels.

Median indoor PM_2.5_ exposure levels by season were 5.9 µg/m^3^ in summer and 7.5 µg/m^3^ in winter. And that of personal levels were 8.0 µg/m^3^ in summer and 9.3 µg/m^3^ in winter, respectively. Although there was a statistically significant difference among seasons, the degree of seasonal difference (ratio of winter to summer: 1.27, 1.16) was relatively smaller in indoor or personal PM_2.5_ compared to that of outdoor PM_2.5_ (ratio of winter to summer: 2.16).

The median seasonal IO ratio (indoor level, compared to outdoor level) in this study ranged from 0.30 to 0.51 in fall, winter, and spring, but its value of summer was 0.70. A study conducted in Beijing, China [17] reported that the median IO ratio was 0.64 ± 0.31 in the fuel-burning season (winter) and 0.81 ± 0.62 in the nonfuel-burning season (summer).

In addition, an IO ratio of 0.88 ± 0.56 in the nonfuel-burning period and 0.53 ± 0.35 in the fuel-burning period was reported [18]. Also reported was a similar trend in 2020 with slopes of 0.172, 0.501, 0.168, and 0.135 for regression equations (x: outdoor, y: indoor) by using outdoor and indoor measurements from spring, summer, fall, and winter, respectively, at several universities in Southeast China [19].

Our study results are consistent with previous study outcomes; the IO ratio in the nonfuel-burning period is higher than that in the fuel-burning period. It is considered that the difference in IO between the nonfuel-burning (summer) and fuel-burning (winter) periods was due to the lower concentration of outdoor PM_2.5_ in the summer than in the winter period while indoor PM_2.5_ concentration levels remained relatively consistent. Deng reported that in Beijing in 2016, the IO was 0.36 when the outdoor fine particulate matter concentration was above 150 µg/m^3^, and the IO was 1.1 when it was below 100 µg/m^3^ [20]. In addition, data reported from China in 2018 showed that when the outdoor PM_2.5_ level was higher than 150 µg/m^3^, the median IO ratio was 0.60 to 0.75 in 46 naturally ventilated homes but was 0.88–0.97 when the outdoor PM_2.5_ level was lower than 75 µg/m^3^, which was also consistent with our study results [21]. Thus, those previous study outcomes and our data indicated that the seasonality of outdoor PM_2.5_ levels could affect the IO ratio because indoor PM_2.5_ levels were relatively consistent.

Our study also demonstrated that PO is approximately 1.3 to 1.5 times higher than IO (Table 2). Our results demonstrated that, compared with real world exposure levels, a person’s exposure level is possibly overestimated when only outdoor PM_2.5_ data are used, while it can be underestimated when only indoor PM_2.5_ data are used. Therefore, for an accurate estimation of a person’s exposure level to PM_2.5_, it is highly recommended to measure personal PM_2.5_ exposure levels. If personal monitoring is unavailable, measuring both indoor and outdoor PM_2.5_ levels and obtaining information of time activity patterns and infiltration rates as well as PM_2.5_ components are recommended [6].

As technology develops, sensor-based personal PM_2.5_ monitoring has become common [7,8,9]. Due to our finding of the degree of differences in indoor exposure level compared to personal exposure level to PM_2.5_ by season, this study could highlight that person-specific mitigation strategies are necessary for the elderly to improve their health outcomes in the future.

However, our study results should be read with care. To achieve our study goal successfully, we selected three different sensors (outdoor: PurpleAir; indoor: Awair; personal: PICO). Due to the typical variables in each sensor that we needed (outdoor: air pressure; indoor: CO_2_; personal: longitude, latitude), we couldn’t but use different sensors. To overcome this problem, we conducted sensor comparison tests two times during the study period, and results showed that there was no sensor malfunction or sensor response problem. Also, as we mentioned in our methods, this study recruited Korean community-dwelling persons of 65 years and older via several university-affiliated hospitals. Therefore, we only examined the exposure levels of this age group. Given that the indoor activities or lifestyles of older individuals may vary from those of the general population, our findings might not be generalized.

Overall, our study has revealed that the real-world exposure level of our study participants to PM_2.5_ was lower than what was initially expected based on the measurements of outdoor data. In other words, we anticipated that the PM_2.5_ exposure experienced by our participants would be higher, given the outdoor measurements of PM_2.5_ levels. However, this study’s results indicated that actual exposure levels to PM_2.5_ are lower than what was predicted based on the outdoor data.

Our findings could have several implications and may prompt further investigation into the reasons for the discrepancy between indoor, personal, and outdoor PM_2.5_ levels. Factors such as indoor air quality, personal behavior patterns, and the effectiveness of any protective measures (e.g., using air purifiers) could contribute to the observed differences.

Understanding the reasons behind the lower PM_2.5_ exposure levels in real-world scenarios is crucial for interpreting this study’s results accurately and determining their significance for public health or environmental considerations. It may also guide future research in this area and inform policies aimed at reducing PM_2.5_ exposure and its potential health impacts.

## 5. Conclusions

In this study, we compared the distribution of PM_2.5_ measured via indoor, outdoor, and personal sampling over four seasons and three different study areas simultaneously. We found that outdoor PM_2.5_ concentrations were different between seasons, especially between summer and winter. Our study also demonstrated that the distribution was different in areas within one season. Also, our study revealed that outdoor PM_2.5_ levels were higher than indoor or personal PM_2.5_ levels, indicating that the real-world exposure level of our participants to PM_2.5_ was lower than what was initially expected based on the outdoor data. Further investigation for elderly-specific populations may need to identify reasons for the discrepancy between indoor, personal, and outdoor PM_2.5_ levels including their personal behavior patterns and the effectiveness of any protective measures (e.g., using air purifiers).

## Figures and Tables

**Figure 1 ijerph-20-06684-f001:**
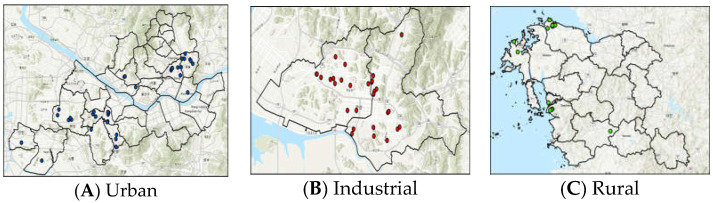
Spatial distribution of study participants recruited at Seoul ((**A**), urban area), Ansan ((**B**), industrial area) and Chungcheongnam-do ((**C**), rural area).

**Figure 2 ijerph-20-06684-f002:**
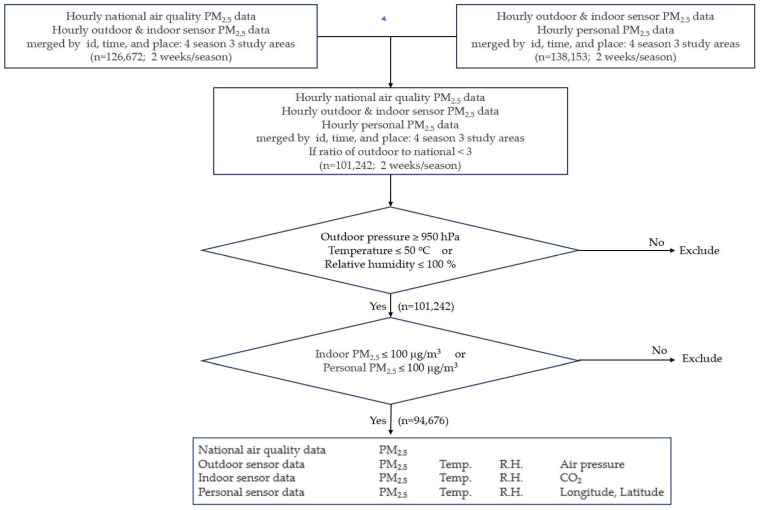
Summary of the data preparation procedure and variables collected from each sensor.

**Figure 3 ijerph-20-06684-f003:**
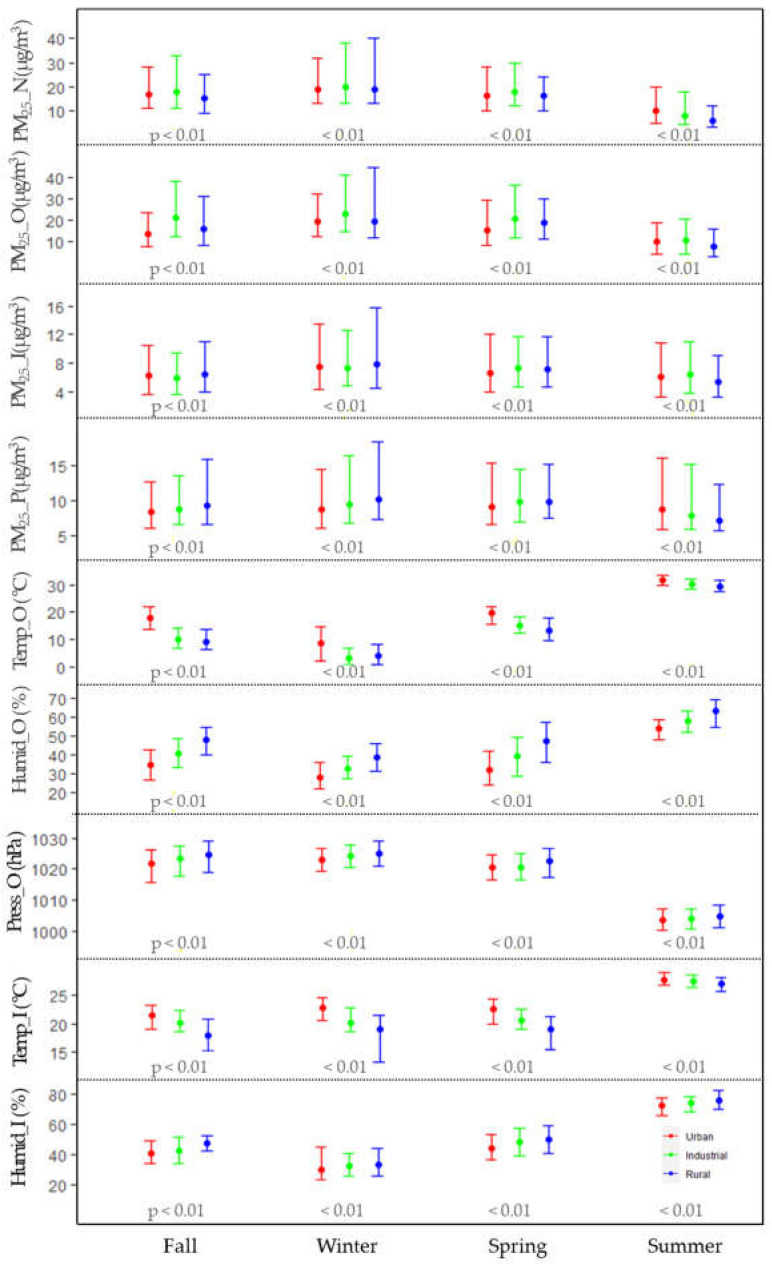
Median (IQR) level of PM_2.5_, temperature, relative humidity, and pressure measured and national site and real-time sensors by season and area (I: indoor, P: personal, O: outdoor, N: National). Each *p*-value was obtained from Kruskal–Wallis test showing the statistical difference of distribution among three areas (Urban: left, Industrial: center, Rural: right).

**Table 1 ijerph-20-06684-t001:** Median (IQR) level of PM_2.5_, temperature, relative humidity, and pressure measured and national site and real-time sensors.

	Overall	Fall	Winter	Spring	Summer	*p*-Value **^,b^
	(n = 94,676 P = 51,866) ^a^	(n = 25,427 P = 13,947)	(n = 25,096 P = 13,679)	(n = 22,923 P = 12,303)	(n = 21,230 P = 11,937)	
	Median	Median	Median	Median	Median	(F:W:Sp:S)
	(Q1–Q3)	(Q1–Q3)	(Q1–Q3)	(Q1–Q3)	(Q1–Q3)	
PM_2.5__N * (μg/m^3^)	16.0	17.0	20.0	16.0	8.0	<0.001
	(9.0–27.0)	(10.0–28.0)	(13.0–28.0)	(10.0–27.0)	(4.0–17.0)	
PM_2.5__O * (μg/m^3^)	16.3	16.4	20.5	18.2	9.5	<0.001
	(9.1–29.4)	(9.1–29.6)	(13.0–38.0)	(10.2–31.8)	(3.8–18.7)	
PM_2.5__I * (μg/m^3^)	6.7	6.2	7.5	7.1	5.9	<0.001
	(4.0–11.5)	(3.7–10.4)	(4.6–13.7)	(4.4–11.8)	(3.4–10.4)	
PM_2.5__P * (μg/m^3^)	8.9	8.7	9.3	9.4	8.0	<0.001
	(6.3–14.7)	(6.3–13.4)	(6.7–16.0)	(6.9–15.0)	(5.7–14.8)	
Temp_O (°C)	15	13.2	5.0	16.1	30.5	<0.001
	(8.0–25.5)	(8.4–18.2)	(0.7–10.8)	(12.2–20.2)	(28.6–32.6)	
RH_O (%)	41.2	39.9	32.4	38.5	57.2	<0.001
	(30.0–52.5)	(30.9–48.2)	(25.0–40.4)	(27.3–49.4)	(50.4–63.8)	
Press_O (hPA)	1020.3	1023.1	1023.8	1021.3	1004.2	<0.001
	(1011.1–1025.7)	(1017.1–1027.5)	(1020.1–1027.7)	(1016.6–1025.4)	(1000.6–1007.4)	
Temp_I (°C)	22.1	20.3	21.2	20.9	27.6	<0.001
	(19.0–25.5)	(17.9–22.4)	(18.0–23.7)	(18.6–23.2)	(26.4–28.6)	
Humid_I (%)	46.6	43.1	31.8	46.9	73.8	<0.001
	(34.7–62.7)	(35.4–51.0)	(24.2–43.4)	(38.6–56.2)	(67.7–78.8)	

* N: National, I: Indoor, O: Outdoor, P: Personal. ** Kruskal–Wallis test. ^a^ “n” is the combined indoor and outdoor sample size, “P” represents the sample size that includes personal measurement data as well as simultaneously measured indoor and outdoor data. ^b^ Bonferroni correction: all PM_2.5_ data distributions were different between two seasons except ‘PM_2.5__N Fall–Spring’, ‘PM_2.5__P Fall–Summer’.

**Table 2 ijerph-20-06684-t002:** Median (IQR) level of I/N, P/N, I/O, P/O by season and study area.

	Urban	Industrial	Rural	*p*-Value *
Fall				
I/N	0.37 (0.23–0.62)	0.33 (0.22–0.51)	0.45 (0.29–0.78)	<0.001
P/N	0.56 (0.40–0.83)	0.52 (0.36–0.78)	0.68 (0.48–1.10)	<0.001
I/O	0.51 (0.28–0.73)	0.30 (0.18–0.47)	0.41 (0.26–0.66)	<0.001
P/O	0.71 (0.45–1.06)	0.46 (0.31–0.68)	0.59 (0.39–0.96)	<0.001
Winter				
I/N	0.37 (0.25–0.61)	0.37 (0.27–0.50)	0.38 (0.27–0.56)	<0.001
P/N	0.52 (0.36–0.72)	0.51 (0.36–0.71)	0.58 (0.42–0.82)	<0.001
I/O	0.42 (0.24–0.63)	0.34 (0.24–0.43)	0.36 (0.26–0.53)	<0.001
P/O	0.50 (0.33–0.68)	0.46 (0.33–0.62)	0.54 (0.39–0.73)	<0.001
Spring				
I/N	0.44 (0.28–0.70)	0.40 (0.26–0.66)	0.50 (0.31–0.85)	<0.001
P/N	0.67 (0.46–1.04)	0.57 (0.38–0.93)	0.80 (0.57–1.25)	<0.001
I/O	0.49 (0.28–0.73)	0.37 (0.23–0.59)	0.42 (0.28–0.62)	<0.001
P/O	0.68 (0.41–1.02)	0.55 (0.33–0.84)	0.63 (0.45–0.86)	<0.001
Summer				
I/N	0.65 (0.39–1.10)	0.81 (0.54–1.32)	0.87 (0.50–1.56)	<0.001
P/N	1.05 (0.74–1.73)	1.14 (0.78–1.88)	1.40 (0.81–2.47)	<0.001
I/O	0.68 (0.48–0.97)	0.70 (0.51–1.01)	0.72 (0.49–1.28)	<0.001
P/O	1.11 (0.84–1.67)	1.07 (0.74–1.57)	1.12 (0.75–2.06)	<0.001

* Kruskal–Wallis test.

## Data Availability

The data are not publicly available due to privacy and ethical restrictions.

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
