# Peer review of "Comparison of Personal or Indoor PM2.5 Exposure Level to That of Outdoor: Over Four Seasons in Selected Urban, Industrial, and Rural Areas of South Korea: (K-IOP Study)"

_ijerph, 2023, doi:10.3390/ijerph20176684_

Round 1
Reviewer 1 Report
The objective of this study was to compare the distribution of indoor, outdoor, and personal PM2.5 concentrations measured simultaneously among participants living in urban, industrial, or rural areas over 4 seasons from November 2021 to July 2022 in South Korea. However, there are many ambiguous points that authors need to consider with careful discussion. Please see the comments below.
1. In abstract, the results should be reported in coincidence with the study topic and objective. In particular, distribution of PM2.5 or I/O or P/O ratios should be reported for urban, industrial, and rural areas separately.
2. In abstract, what is the meaning of p-value < 0.05? This should be clearly specified.
3. References for the statement “an increasing number of studies are focusing on indoor air pollution levels.” in line 46 (Page 2) are needed to make this information stronger.
4. In section 2.1, the clear reason why only elderly was considered as study participants since saying that elderly people are one of the most vulnerable groups to PM2.5 exposure is not strong enough, otherwise findings from this study cannot be generalized to general populations.
5. Line 91 (Page 3) Should “The content form” be “The consent form”?
6. Lines 104-105, the sentence “The indoor, outdoor, and personal” has not completed yet, so that it should be rewritten.
7. Lines 109-116 (Page 3), indoor sensor was measured every 5 minutes, outdoor every 2 minutes and personal sensor every 1 minute. The measurement intervals are different among indoor, outdoor, and personal that may be difficult to compared among another. Why not measured at the same interval? This needs a clear explanation.
8. Line 155, authors mentioned that “Outdoor PM2.5 values were excluded from the analysis when outdoor temperature was above 65°F (40°C)”, which is different from the information in Figure 2 (<=50°C). Please check this again carefully which one is correct and revise accordingly.
9. Lines 184-185, authors explained in Table 1 that “In summer, the temperature and humidity were the same indoors and outdoors”.I don’t agree with this explanation because temperature for indoor vs outdoor was 27.6 (26.4-28.6) vs 30.5 (28.6-32.6) and humidity for indoor vs outdoor was 73.8 (67.7-78.8) vs 57.2 (50.4-63.8). You can see that the values of IQR range for indoor and outdoor temperature and humidity were not even overlapped, suggesting that indoor and outdoor temperature and humidity in summer was about significantly different. Please check this explanation again carefully.
10. In Table 1, what is the meaning of “n” and “P” that are mentioned in the parenthesis for overall, fall, winter, spring, and summer? These should be clearly defined.
11. For validation of PM2.5 concentration obtained from light scattering sensors, could authors please show whether outdoor PM2.5 concentration from sensors is well correlated with that from monitoring stations in scatter plot with validated parameters (R, R^2, RMSE, etc.)? Moreover, as authors applied three different sensors, the correlation plot among different types of sensors is needed.
12. To be honest, Figure S1 is difficult to understand, where the meaning of different colors was not specified in the plot.
13. Lines 200-246, the explanation is very confusing. Specifically, there are a lot of numerical values without showing those values in specific Table. In particular, I personally cannot capture what did authors mean by “regional difference”. In this case, authors need to put numerical data for all regions in specific table and show the results of statistical test for the difference. Showing data in Figure and explain those in number would be hard to understand, unless authors add supporting tables in supplementary information. The same comments for what authors explained in lines 252-263.
14. Lines 288-289, the authors explained that “IO ratio in the non-heating period is higher than that in the heating period.”, but the reported results are contradicted with this explanation, where IO in summer (maybe heating period) was 0.70 was higher than that in fall, winter and spring which is ranged from 0.30 to 0.51 as authors showed in line 281. Please discuss your finding with care.
15. Conclusion did not follow the study’s objective. Specifically, the study’s objective was “to compare distribution of indoor, outdoor, and personal PM2.5 concentrations measured simultaneously among participants living in urban, industrial, or rural areas over 4 seasons from November 2021 to July 2022”, where the authors only compared personal and outdoor PM2.5 without concluding on different areas and seasons. They need to be careful on this concern.
I found some grammatical errors that should be carefully checked again throughout the manuscript. For example, line 298 “ those previous study outcome”, where “outcomes” should has “s”.
Author Response
The authors thank all reviewers for their suggestions and the time and effort they have invested in reviewing the manuscript. We appreciate your insightful comments that have contributed to the improvement of the clarity and quality of the manuscript. The responses to their comments have been prepared and are given below.

Reviewer 2 Report
Dear Authors,
Many thanks for the chance to go through your manuscript.
To help improve the quality consider the following points;
i. How long did the participant attach the sensor to themselves?
ii. Also was there any relationship considered between temperature/wind speed... and PM2.5 distribution I/O?
iii. was there any consideration/observation regards the activities undertaken indoor/outdoor and PM2.5 distribution? This was not reported in the methodology section.
iv. Earlier inline 87-88 there was a statement that reads "....elderly people who were able to install indoor and outdoor sensors in their homes". Does that mean they were charged with this responsibility? and if so how sure that position of each sensor did meet the 1.5 meter height stated in line 191 and setting up of the instrument that led to the data collected remotely? Authors need to clarify this further.
iv. Authors need also to consider the limitation of the study moving forward
Author Response
- How long did the participant attach the sensor to themselves?
Response:
To better describe our data collection procedure, we have included this information in the methods:
(Before)
The personal sensor was carried in a carrying pouch and it was attached to the subjects’ waist belt or bag, and conducted measurements at 1 min intervals.
(After)
Participants who agreed to personal sampling carried the personal sensor in a carrying pouch attached to the participant’s waist belt or bag, and measurements were obtained at one-minute intervals over two weeks per season or the longest period spanning at least five consecutive days.
- Also was there any relationship considered between temperature/wind speed... and PM2.5 distribution I/O?
Response:
Identifying determinant factors for the IO was not the objective of this study. This will be investigated in our upcoming paper. We have included this issue as a recommendation for future research:
“Further studies on older individuals should investigate the reasons for the discrepancy between indoor, personal, and outdoor PM2.5 exposure levels, including individual behavior patterns and the effectiveness of any protective measures (e.g., using air purifiers) or meteorological conditions.”
- was there any consideration/observation regards the activities undertaken indoor/outdoor and PM2.5 distribution? This was not reported in the methodology section.
Response:
To clarify indoor and outdoor activities recorded in diaries, we have added the following description to the methods:
(Before)
The diary content of the temporal activity pattern collected during the measurement period was also linked to the hourly data.
(After)
“The diary information on temporal activity patterns (1. sleeping at home, 2. resting at home, 3. socializing outside the community center, 4. socializing outside shops/restaurants, 5. exercise, 6. cleaning indoors, 7. farming outdoors, 8. cooking indoors, 9. working outdoors as a part-time job) collected during the measurement period was also linked to the hourly data.” Line 144)
“Further studies on older individuals should investigate the reasons for the discrepancy between indoor, personal, and outdoor PM2.5 exposure levels, including individual behavior patterns and the effectiveness of any protective measures (e.g., using air purifiers) or meteorological conditions.” (Line 366)
- Earlier inline 87-88 there was a statement that reads "....elderly people who were able to install indoor and outdoor sensors in their homes". Does that mean they were charged with this responsibility? and if so how sure that position of each sensor did meet the 1.5 meter height stated in line 191 and setting up of the instrument that led to the data collected remotely? Authors need to clarify this further.
Response:
To clarify our methods, we have revised the sentence and included information on sensor installation as follows:
(Before)
The study subjects were elderly people who were able to install indoor and outdoor PM2.5 sensors in their homes.
(After)
“The study participants were older individuals who were able to install indoor and outdoor PM2.5 sensors in their homes. We selected volunteers who could install a multi-tap sensor outside their houses or on verandas or terraces that allow adequate ambient air circulation. Simultaneously, the indoor sensors were installed in living rooms at a height of approximately 1.5 m from the floor.
Field managers visited participants’ houses at least twice per season (at the start and end of seasonal sampling or when requested).
We paid compensation fees each season at the end of seasonal sampling. The fees were paid four times in total.”
- Authors need also to consider the limitation of the study moving forward
Response:
As suggested, we have included more information on the limitations of our study:
(Line 338)
“Our study results should be interpreted with care. To achieve our study objectives successfully, we selected three different sensors (outdoor: PurpleAir; indoor: Awair, personal: Pico). Due to the variables each sensor was required to measure (outdoor: air pressure; indoor: CO2 levels; personal: longitude, latitude), different sensors had to be used. To overcome this problem, we conducted sensor comparison tests twice during the study period, and the results indicated no sensor malfunction or response problems.” (Supplementary information)
“Additionally, this study design recruited Korean community-dwelling persons of 65 years and older via several university-affiliated hospitals. Therefore, we only examined the exposure levels of this age group. Because the indoor activities or lifestyles of older individuals may vary from that of the general population, our findings should not be generalized. However, in modern society, most people spend more time indoors than outdoors. Although specific activity patterns are different between younger and older people, older individuals are exposed to indoor sources more often than outdoor sources. Therefore, our finding that the real-world PM2.5 exposure level of our study participants was lower than what was initially expected based on the outdoor data remains valid.”
“Further studies on older individuals should investigate the reasons for the discrepancy between indoor, personal, and outdoor PM2.5 exposure levels, including individual behavior patterns and the effectiveness of any protective measures (e.g., using air purifiers) or meteorological conditions.”

Reviewer 3 Report
Very interesting research.
- line 105 - missing the end of the sentence
- Figure 2 - can you please put 2.5 of PM2.5 and 2 of CO2 in subscript also in the figure?
- line 160 - put a space between finally and 94676 data
- line 162 - what about autumn?
- paragraph 3.2 = figure 3 - don't duplicate the same results
- line 294 - misspelled mass concentration 150 ug/m3
- line 310 - dot after references is red instead of black color
- Conclusions - could you please expand the conclusions
Moderate or even minor editing of English language required
Author Response
Very interesting research.
- line 105 - missing the end of the sentence
Response:
To rectify this, we have revised the sentence as follows:
(Before)
The indoor, outdoor, and personal
All PM2.5 sensor devices used in this study were certified as class 1 with an error rate of less than 20% with comparison of national reference methods system
(After)
“The indoor, outdoor, and personal PM2.5 sensor devices used in this study are certified as class 1, with an error rate of less than 20% compared with the national reference methods system.”
- Figure 2 - can you please put 2.5 of PM2.5 and 2 of CO2 in subscript also in the figure?
Response:
As suggested, we have revised Figure 2 as follows: 2.5 of PM2.5 and 2 of CO2 in subscript.
- line 160 - put a space between finally and 94676 data
Response:
As suggested, we have inserted a space between “Finally” and “94,676” in line 160.
- line 162 - what about autumn?
Response:
To maintain consistency, we decided to retain the use of “fall”, which is used in American English as a synonym for “autumn”, throughout the manuscript since we have used the term “fall” in all our tables, figures, and text, We thank the reviewer for the suggestion, nonetheless.
- paragraph 3.2 = figure 3 - don't duplicate the same results
Response:
Paragraph 3.2 provides a summary of Figure 3. As suggested, we have shortened the paragraph by removing duplicated results.
- line 294 – misspelled mass concentration 150 ug/m3
Response:
We have corrected the error in the unit.
- line 310 - dot after references is red instead of black color
Response:
We have rechecked and corrected the color of the dot.
- Conclusions - could you please expand the conclusions
Response:
We have included the following information in the supplemental information to elaborate on our conclusions:
“In this study, we compared the distribution of PM2.5 measured via indoor, outdoor, and personal sampling over four seasons and three different study areas simultaneously. We found that outdoor PM2.5 concentrations were different between seasons, especially between summer and winter. Our study also demonstrated that the distribution was different in areas within one season. Also, our study revealed that outdoor PM2.5 levels were higher than indoor or personal PM2.5 levels, indicating that the real-world exposure level of our participants to PM2.5 was lower than what was initially expected based on the outdoor data. Further studies on older individuals should investigate the reasons for the discrepancy between indoor, personal, and outdoor PM2.5 exposure levels, including individual behavior patterns and the effectiveness of any protective measures (e.g., using air purifiers).”

Round 2
Reviewer 1 Report
Thank you for your effort addressing my concerns. I have no further comment.